# Simultaneous Free Fibula and Anterolateral Thigh Flap in Lower Extremity Reconstruction Following Osteomyelitis in a Trauma Patient: A Case Report

**DOI:** 10.3390/medicina59071206

**Published:** 2023-06-27

**Authors:** Tadej Voljc, Michael Schintler, Anna Vasilyeva, Lars-Peter Kamolz, Heinz Buerger

**Affiliations:** 1Division of Plastic, Aesthetic and Reconstructive Surgery, Department of Surgery, University Hospital Graz, 8036 Graz, Austria; 2Division of Hand and Microsurgery, Private Hospital Maria Hilf, 9020 Klagenfurt am Woerthersee, Austria

**Keywords:** bone, case report, free tissue flaps, infection, lower extremity, microsurgery

## Abstract

This case report focuses on a 17-year-old polytrauma patient who suffered a septic wound infection after an open reduction and internal fixation (ORIF) and soft tissue reconstruction with a pedicled flap, which led to a substantial bone and soft tissue defect of the lower leg. After thorough antibiotic treatment and after ensuring a non-septic wound, the defect was reconstructed using a contralateral free fibula flap designed as a flow through flap in a double loop manner to accommodate two fibular fragments and an ipsilateral ALT flap. Early weight bearing was initiated 11 days after the free flap transfer under external fixation, with full weight bearing achieved in 36 days with external fixation. After the removal of external fixation, full weight bearing was able to be reinitiated after 13 days, leading to the patient’s return to normal activity 6 months after the bony reconstruction. This case presents an innovative approach to treating a complex defect, with the final decision on using two separate free flaps instead of a single osteofasciocutaneous free flap resulting in a good bony reconstruction and soft tissue coverage, and with the use of external fixation enabling early rehabilitation.

## 1. Introduction

Treating traumatic defects of weight-bearing bone in children and young active adults can prove challenging, especially after a local or systemic infection. In the following case, we share our experience of successfully treating a complex defect through a microsurgical tissue transfer, after the failure of initial treatment using open reposition and internal fixation (ORIF) together with local soft tissue coverage due to a local wound and bone infection, and sepsis.

## 2. Case Description

Our patient is a 17-year-old male competitive athlete who suffered a traffic accident riding a motorcycle, and he had no relevant preexisting medical conditions or interventions. He sustained a comminuted open left proximal tibial fracture with articular involvement (Figure 1), an injury to the medial meniscus and the medial collateral ligament, and a soft tissue defect of 10 × 7 cm of the proximal anteromedial aspect of the left lower leg. Furthermore, he sustained an impacted right femoral fracture, a multifragmentary fracture of the right proximal tibia without articular involvement (Figure 2), a C7 spinal process fracture, and a soft tissue laceration of the right elbow.

The left extremity has initially been fixed with an external fixator, and the right femoral and the tibial fractures were initially treated with intramedullary nailing. A compartment release of the right lower leg was required on day 3 after admission. Following necrectomy, an internal fixation of the left tibia was performed on day 8 after injury with a locking compression plate along with closure of the soft tissue defect with a pedicled medial gastrocnemius muscle flap and a split-thickness skin graft. The meniscus and the med. collateral ligament were reconstructed. The immediate postoperative period was uneventful; however, a continuous wound discharge from the right lower leg 2 weeks after the compartment release led to the examination of microbial cultures, with *P. aeruginosa* being isolated from the wounds of the right and left lower leg as well as the right elbow.

Upon admission, the patient received a prophylactic antibiotic therapy of Cefuroxime and Clindamycin, which was changed to Piperacillin and Tazobactam 2 weeks after admission. On the isolation of *P. aeruginosa*, the therapy with Cefuroxime and Clindamycin was reinitiated alongside Piperacillin and Tazobactam. On the left lower leg, a hematoma has developed under the med. gastrocnemius muscle flap, which was evacuated, and an antibiotic bead chain as well as Palacos bone cement were inserted, both containing Gentamycin as a local antibiotic agent. However, 11 days after the confirmed infection, the patient developed septic shock with a blood leucocyte count of 2.54 × 10^9^/L, CRP value of 282.4 mg/L and blood Hb of 9.2 g/dL. A plate removal from the left tibia was required. Following the plate removal, a steady improvement in blood inflammatory markers was noted, despite microbial cultures from the plate itself remaining negative. Laboratory findings 5 days after the plate removal showed a blood leucocyte count of 8.49 × 10^9^/L, a CRP value of 18.0 mg/L, and a blood Hb of 10.3 g/dL. The right leg was successfully treated with necrectomy and continued V.A.C. therapy, and secondary closure was able to be attained 2 weeks after the confirmed infection.

The patient was discharged from hospital care 3 weeks after the confirmed *P. aeruginosa* infection under oral Ciprofloxacin therapy without any local or systemic signs of infection. No signs of infection whatsoever were noted on his visits to the outpatient clinic for another 5 weeks, allowing for a secondary reconstructive effort on the left lower leg. A stable weight-bearing bone construct and an adequate amount of well-perfused soft tissue to fill the defect left behind by the infection were identified as the main requirements, and so a reconstruction with a contralateral free fibula flap and an ipsilateral anterolateral thigh (ALT) flap was chosen. Figure 3 shows the left lower leg prior to the free flap transfer.

The free fibula flap was attached to the post. tibial vessels as a flow-through flap in a double-loop manner with two venous anastomoses to accommodate two separate fibular fragments. The ALT flap was attached to the fibular vessels of the free fibula flap (see Figure 4). Care was taken to minimize the amount of osteosynthetic material used during osseous fixation of the free fibula flap in order to reduce the risk of further infection, and no more than three screws were used to fix both of the fibular fragments. After 11 days, external fixation was applied and the patient was allowed to increase weight bearing, with full weight bearing achieved after 36 days under external fixation. Increasing mobilization and weight bearing were well tolerated and good osseous remodeling was observed, and so the external fixation was able to be removed after 14 weeks. After removal, the patient was restricted to partial weight-bearing for 13 days, followed by a return to full weight-bearing without external fixation. A total of 6 months after the free flap transfer and 10 months after the injury, the patient was again able to participate in sports.

## 3. Discussion

Reconstruction after injury or limb-sparing oncologic resection in the pediatric population can be challenging. After an initial period of caution regarding both reliability and feasibility, microsurgical reconstruction has, with advances in operative and postoperative techniques, gained widespread acceptance in this field [1,2]. Since its introduction in the mid-1970s, the free fibula flap has been an extensively used flap for lower and upper extremity reconstruction when the bridging of long bone defects is required [3,4]. It is particularly useful for the reconstruction of weight-bearing bone due to its ability to remodel and hypertrophy in response to mechanical stress, and it shows comparable results in regards to primary union, complication, and reintervention rates when compared to alternative techniques, such as bone allo- and autografting, bone transport, and endoprosthesis, in a wide patient population [5,6,7]. While subjective donor site morbidity remains low following free fibula transfers, reautomatization and restoration of gait can still be incomplete [8,9,10,11]. Nonetheless, free fibula transfers offer a good treatment option for tibial nonunion following infection [12], possibly, in part, because of the vital bone marrow included in the flap [13]. Designing the free tissue transfer as a flow-through flap preserves perfusion for either tissues distal to the defect, functioning as a bypass, or to supply a second free flap in more complex reconstructive procedures [14,15]. While local as well as free perforator flaps are an important tool in the hands of plastic surgeons, the ALT flap, described in 1984 by Song et al. remains a workhorse flap for soft tissue reconstruction, despite anatomical variations in its vascular pedicle [16,17,18,19,20,21]. In our experience, the ALT flap provides ample soft tissue and skin coverage for even larger defects. The donor site morbidity of the ALT flap is minimal [16,22]. Patient and case-specific factors remain a critical aspect in deciding on the suitability of the technique when managing long bone defects.

Apart from the general limitations of a case report, this case is limited by focusing on a young healthy patient. The rate of complications, reinterventions and reconstructive failure, as well as the duration of rehabilitation can all change in a different patient population. However, the case does illustrate the potential of a well perfused tissue transfer in addressing a complex bone and soft tissue defect following infection.

Care was taken to provide a clean post-septic wound with viable wound margins prior to attempting a free flap reconstruction, and care was also taken to use a minimal amount of osteosynthetic material. While presented with the option to use a single osteofasciocutaneous free fibula flap, the decision was made to use two separate free flaps—due to scar tissue in the cutaneous portion of the contralateral free fibula flap following prior compartment release, and also because, in comparison, a larger volume of viable and well-perfused soft tissue was able to be harvested using an ALT flap. An ipsilateral flap instead of a contralateral free fibula flap was dismissed as an option in order to avoid any further compromise of stability in the left leg. Since gait restoration of the donor leg after a free fibula transfer has, as mentioned above, been shown to be incomplete, a compelling area of further research would be a computational simulation approach to model gait patterns of donor legs and recipient legs after free fibula transfers, similar to the techniques used for hip joint prosthesis, in order to identify possible focus areas for rehabilitation [23].

At the time of writing, minor surgical scar corrections have been performed, with Figure 5 showing the patient’s condition prior to scar correction. However, with the recent increase in usage modalities and the number of publications on hypertrophic scar management using lasers, laser systems warrant increased attention in the field of scar management and correction [24,25,26,27].

## 4. Conclusions

A complex lower limb defect, as described above, can be treated through microsurgical reconstruction when first line treatment methods fail, so long as a clean non-septic defect can be obtained. The decision to use two separate free flaps seems, in hindsight, to be crucial, as with this approach, both a good osseous reconstruction and ample well-perfused soft tissue to fill the defect were able to be produced, and this enabled both early weight-bearing and the further reduction of the risk of infection whilst also providing coverage.

A good functional and cosmetic result was able to be achieved and, despite a minimal amount of internal osteosynthesis, early weight-bearing was possible under external fixation. This led to early weight bearing without external fixation, andt provided the patient with a rapid return to normal activity.

## Figures and Tables

**Figure 1 medicina-59-01206-f001:**
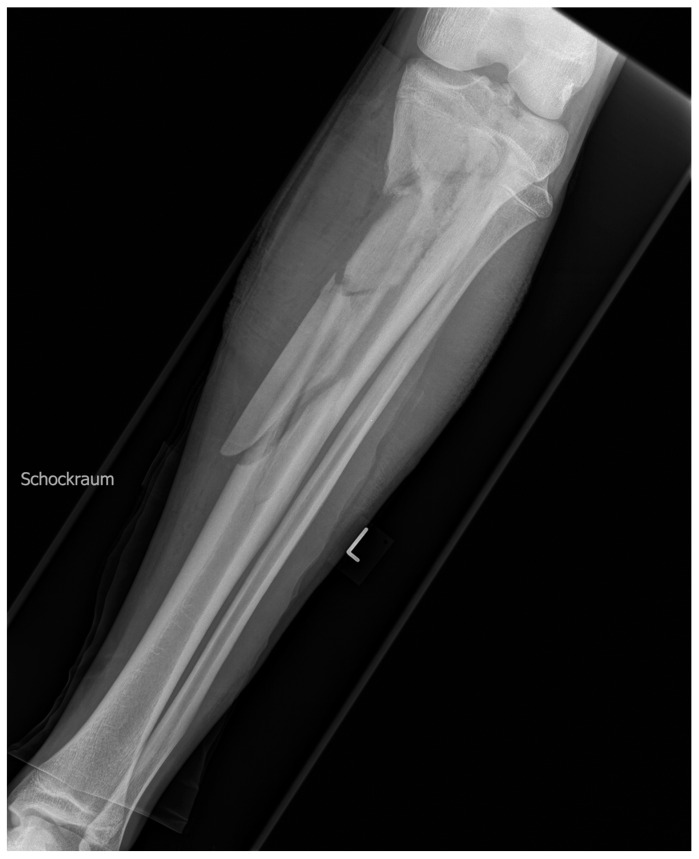
X-ray of the left lower leg on admission.

**Figure 2 medicina-59-01206-f002:**
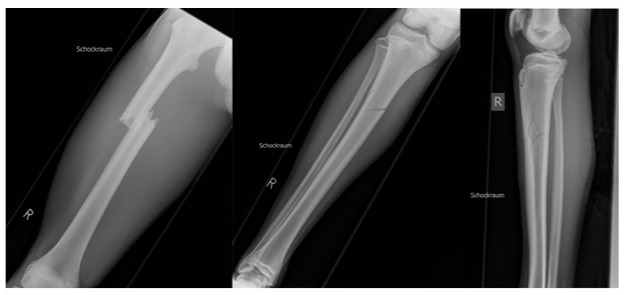
X-ray of the right lower extremity on admission. From left to right: right upper leg, right lower leg.

**Figure 3 medicina-59-01206-f003:**
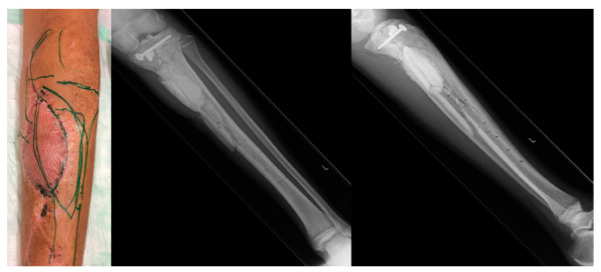
Left lower leg prior to the free flap transfer: From left to right: visible med. gastrocnemius flap with the split-thickness skin graft and markings as part of the preoperative planning; X-ray of the left lower leg prior to the free flap transfer.

**Figure 4 medicina-59-01206-f004:**
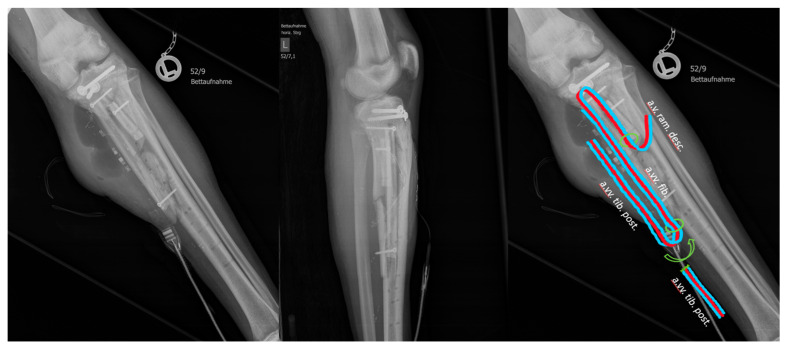
X-ray of the left lower leg after the free flap transfer, with a schematic of the relevant vasculature (artery red, veins blue, anastomoses marked with circles).

**Figure 5 medicina-59-01206-f005:**
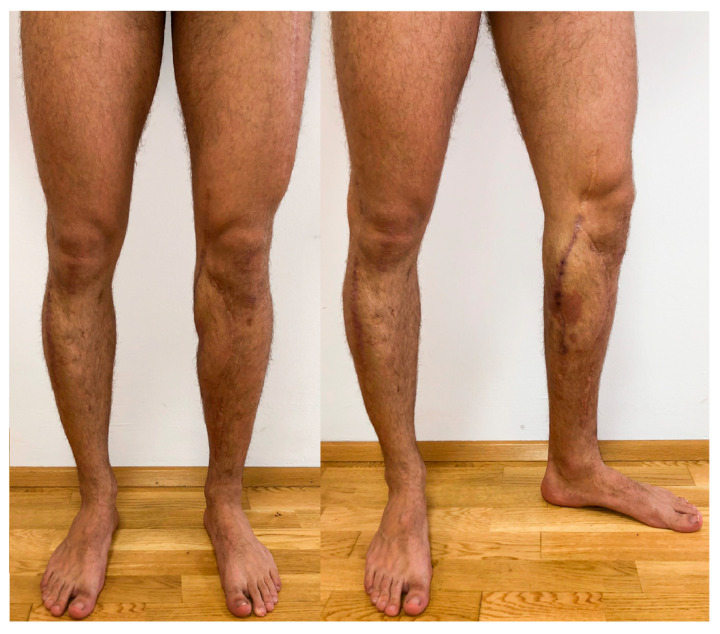
Both lower extremities 53 months after the free flap transfer.

## Data Availability

Data available on request through the corresponding author.

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
