# Peer review of "Simultaneous Free Fibula and Anterolateral Thigh Flap in Lower Extremity Reconstruction Following Osteomyelitis in a Trauma Patient: A Case Report"

_medicina, 2023, doi:10.3390/medicina59071206_

Round 1
Reviewer 1 Report
- in the case of multiple fractured patients, in which multiple reconstruction surgeries are required, there is always a risk of infection that can lead to osteomyelitis
- in some cases, even if we take all the necessary methods in order to lower the rate of septic complications, infection does occur, which can prolong the hospitalization time and increase the time needed for rehabilitation
- the single stage option of free fibula and anterolateral thigh flap following local infection is a good option, providing coverage of the bone tissue; the fact that they used VAC therapy before the surgical procedure increases the chance of success with this procedure.
- it would have been relevant to prevent also the levels of inflammatory markers, hemoglobin, leucocytes
- the patient's history is presented in an easy-to-understand way, pinpointing the more important information;
- all stages of the treatment were presented with the allocated time frame, the information is easy to read and understand
- the bibliography cited, although is not very long, is in direct relation with the surgical procedure and up to date with the current trends
The English language presented is good, only requiring minor editing.
Author Response
Dear Sir or Madam,
I would like to extend sincere thanks from our entire team for your constructive comments.
We hope you will find the corrected version of the manuscript, with now stated laboratory values from the period of septic shock and an extended Discussion and Reference list, a sufficient improvement.
The following corrections were incorporated into our manuscript using the “Track Changes” function in Microsoft Word:
- Laboratory findings of inflammatory markers were included (page 3)
- The Discussion has been extended (page 5):
- adding the mention of the timeframe of the first description of the free fibula flap,
- highlighting the ability of the vascularized fibula flap to hypertrophy when exposed to mechanical stress,
- stating the donor site morbidity of the free fibula and ALT flaps,
- discussing the role of the free fibula flap in managing nonunion of the tibia due to infection
- mentioning the versatility of flow-through free flaps.
- The reference list was expanded.
- The manuscript has been thoroughly checked for English language errors and corrections were applied.
Should you have any further comments, we would kindly ask you to let us know.
Yours faithfully,
Tadej Voljc
Reviewer 2 Report
I have minor comments to improve the present submitted manuscript overall quality. The manuscript present case reports in perspective of clinical study. It would extend the explanation to discuss the more comprehensive perspective in lower extremity study from computational, experimental, and analytical investigation. The authors invited to read and incorporated the relevant suggested reference as follows: Adopted walking condition for computational simulation approach on bearing of hip joint prosthesis: review over the past 30 years. Heliyon. 2022;8: e12050. doi:10.1016/j.heliyon.2022.e12050
-
Author Response
Dear Sir or Madam,
I would like to extend sincere thanks from our entire team for your constructive comments.
We hope you will find the corrected version of the manuscript, with now stated relevant laboratory values from the period of septic shock and an extended Discussion and reference list, a sufficient improvement.
The following corrections were incorporated into our manuscript using the “Track Changes” function in Microsoft Word:
- Laboratory findings of inflammatory markers were included (page 3)
- The Discussion has been extended (page 5):
- adding the mention of the timeframe of the first description of the free fibula flap,
- highlighting the ability of the vascularized fibula flap to hypertrophy when exposed to mechanical stress,
- stating the donor site morbidity of the free fibula and ALT flaps,
- discussing the role of the free fibula flap in managing nonunion of the tibia due to infection
- mentioning the versatility of flow-through free flaps.
- The reference list has been expanded.
The article suggested by you in the comment (https://www.ncbi.nlm.nih.gov/pmc/articles/PMC9730145/) is a very well-presented review article on a very interesting topic. However, we fail to see it’s parallels with our submitted case report.
We have expanded the Discussion. The suggestion to "discuss the more comprehensive perspective in lower extremity study from computational, experimental, and analytical investigation", perhaps focusing, as per the article suggested, on the mechanical load distribution and bearing on lower extremity free flaps, is a very fine idea for a future clinical study or perhaps a review article, both of which will be strongly considered, yet it seems to reach beyond the scope of a reconstructive case report, such as the one submitted by our team.
Our article has been thoroughly checked for issued in the English language and corrections were applied. If you still see any specific shortcomings regarding the English language, I would kindly ask you to point them out to us.
Thank you.
Yours faithfully,
Tadej Voljc
Round 2
Reviewer 2 Report
The explanation does not incorporate into the revised manuscript along with the literature. It makes the discussion so weak. Please address the comments precisely as listed.
-I am recommending for rejection in the present manuscript.
Author Response
Dear Sir or Madam,
Please find attached the manuscript, revised according to your comments.
The most recent changes in the Discussion and References sections are marked red using the “track changes” function in Microsoft Office Word. Some of the previous corrections, based on the comments of the Academic Editor, have already been incorporated into the text.
We look forward to your reply.
Yours faithfully,
Tadej Voljc
